# Mechanisms Driving the Emergence of Neuronal Hyperexcitability in Fragile X Syndrome

**DOI:** 10.3390/ijms23116315

**Published:** 2022-06-05

**Authors:** Pernille Bülow, Menahem Segal, Gary J. Bassell

**Affiliations:** 1Department of Cell Biology, Emory University School of Medicine, Atlanta, GA 30322, USA; 2Department of Brain Science, Weizmann Institute of Science, Rehovot 76100, Israel; menahem.segal@weizmann.ac.il

**Keywords:** Fragile X Syndrome, the Fragile X Messenger Ribonucleoprotein, FMRP, FMR1, hyperexcitability, homeostatic plasticity, ion channels

## Abstract

Hyperexcitability is a shared neurophysiological phenotype across various genetic neurodevelopmental disorders, including Fragile X syndrome (FXS). Several patient symptoms are associated with hyperexcitability, but a puzzling feature is that their onset is often delayed until their second and third year of life. It remains unclear how and why hyperexcitability emerges in neurodevelopmental disorders. FXS is caused by the loss of FMRP, an RNA-binding protein which has many critical roles including protein synthesis-dependent and independent regulation of ion channels and receptors, as well as global regulation of protein synthesis. Here, we discussed recent literature uncovering novel mechanisms that may drive the progressive onset of hyperexcitability in the FXS brain. We discussed in detail how recent publications have highlighted defects in homeostatic plasticity, providing new insight on the FXS brain and suggest pharmacotherapeutic strategies in FXS and other neurodevelopmental disorders.

## 1. Fragile X Syndrome 

An imbalance in excitatory and inhibitory signaling is a shared neurophysiological phenotype across neurodevelopmental disorders, such as autism, as well as monogenic forms of autism, Fragile X Syndrome (FXS), and Rett Syndrome (RS). This imbalance typically favors exaggerated excitation, leading to dysregulated neuronal signaling, a phenomenon referred to as neuronal hyperexcitability. Neuronal hyperexcitability, particularly in the cortex, has been associated with seizures, sensory hypersensitivity, and sleep disturbances—symptoms that are highly prevalent in various genetic neurodevelopmental disorders [1,2]. A puzzling feature of these symptoms is that their onset is often delayed, such that patients do not express sensory hypersensitivity or seizures until around their second year of life [3]. 

Dysfunction of distinct ion channels or ligand-gated receptors contributes to the expression of hyperexcitability in adult animal models of genetic neurodevelopmental disorders [4,5]. However, it continues to remain unclear how hyperexcitability emerges, and whether an underlying mechanism is shared across disorders. The genetic mutation causing Fragile X Syndrome and its associated symptoms is uniquely well-studied [6], thus offering a rich opportunity to test the mechanisms driving hyperexcitability during development. The overarching question highlighted in this review was how hyperexcitability emerges in Fragile X syndrome, focusing on physiological and molecular mechanisms uncovered in the *Fmr1* knockout mouse model. We first discuss the foundational literature on symptomology and disease genetics of FXS, followed by a review of the most recent literature addressing how FMRP regulates neuronal excitability through ion channels. We then focus on how hyperexcitability is expressed in the FXS brain in a cell-type-specific manner and the consequences for behavior. We follow by asking what drives the emergence and maintenance of abnormal neuronal excitability and plasticity in FXS, which leads to a discussion of the latest research addressing this important topic. Of this literature, we focus particularly on recent studies that have highlighted defects in homeostatic plasticity in FXS and discuss their potential links to the emergence of cortical neuronal hyperexcitability [7,8,9]. We end with a discussion of future directions for the field of FXS and other neurodevelopmental disorders in the pursuit of understanding and ameliorating symptoms associated with neuronal hyperexcitability. This review integrates research from multiple fields, many of which could not be comprehensively discussed here. We refer readers to previous reviews throughout the article for more in depth understanding of each of these. 

### 1.1. Characteristic Symptomology

Fragile X Syndrome (FXS) is a genetic neurodevelopmental disorder that is primarily characterized by intellectual disability [6]. Clinical symptoms include sensory hypersensitivity to tactile, visual, and auditory stimuli, seizures (including epilepsy), and attention deficit hyperactivity disorder [3,4]. The genetic mutation causing FXS is also the most common monogenetic cause of autism—a feature that is consistent with the highly overlapping symptomology, particularly regarding their shared difficulties in emotion regulation and social interactions [10]. However, FXS differs in some important ways from autism. One distinction is that FXS patients display high levels of social anxiety rather than a lack of interest in people, which is a notable feature of children within the autism spectrum disorders. Thus, FXS patients typically demonstrate withdrawal and avoidance from social and/or unfamiliar situations due to anxiety. Unsurprisingly, a large proportion of people with FXS present with co-morbid panic attacks and social anxiety disorders [3]. 

FXS is estimated to affect 1 in 7000 males and 1 in 11,000 females [11]. Approximately 10–20% of people with FXS also suffer from epilepsy (typically representing Benign Focal Epilepsy of Childhood (BFEC)) [12]. Forty-six percent of males with FXS are diagnosed with autism [13]. The majority of FXS patients exhibit symptoms equivalent to Sensory Processing Disorder [3,14]. There is currently no cure for FXS, but people with FXS are often medicated for their co-occurring symptoms (such as epilepsy and ADHD). As discussed below, there have been major efforts to develop new therapeutic agents that can overcome the intellectual and social debilities experienced in FXS, as well as how to ameliorate seizures and sensory hypersensitivities in FXS.

### 1.2. Disease Genetics

FXS is typically caused by a CCG triplet repeat expansion in the Fragile X Messenger Ribonucleoprotein 1 (*FMR1*) gene (previously known as the Fragile X Mental Retardation 1 gene) on the X chromosome [15]. Hypermethylation and transcriptional silencing of the *FMR1* gene leads to loss of the expression of its protein product Fragile X Messenger Ribonucleoprotein (FMRP) (previously known as the Fragile X Mental Retardation Protein) [16]. This X-linked dominant inheritance explains why males are more commonly and more seriously affected than females [6,16]. Premutation carriers can be affected by disorders with different disease mechanisms than FXS that involve RNA toxicity and/or RAN translation, but evidence also suggests haploinsufficiency of FMRP [17]. Fragile X-associated tremor/ataxia syndrome (FXTAS) is a neurodegenerative disorder primarily characterized by ataxia [18]. The premutation allele can also lead to Fragile X-associated primary ovarian insufficiency (FXPOI), which affects ovary function and fertility in women [19]. Haploinsufficiency of FMRP in the premutation range may lead to array of neuropsychiatric impairments, although the underlying mechanisms are not well understood [20].

### 1.3. Characteristic Disease Pathology

A large body of literature has identified disease neuropathology in FXS from both human patients and mouse models. One of the most distinct features is the increased number of immature dendritic spines and/or reduced stability [21,22,23]. These defects are thought to be caused by impaired synaptic proteostasis, leading to faster generation of new synapses but reduced stabilization [24,25,26,27]. While this phenotype has been described extensively, it is unclear how it directly affects behavior. So far, no studies have linked increased dendritic spine number or morphology with neuronal hyperexcitability, and we therefore refer the interested reader to previous reviews on this topic [6,22]. 

## 2. The Fragile X Messenger Ribonucleoprotein—Canonical and Non-Canonical Mechanisms Affecting Ion Channels 

The *FMR1* gene encodes for FMRP, which is an RNA-binding protein that is expressed widely through the body and particularly in the nervous system. FMRP is reported to bind around 4% of the mRNA in the mammalian brain [28], and many of these mRNAs are critical during brain development neuronal plasticity [6,29]. FMRP expression is especially high in cerebral cortex, hippocampus, and cerebellum [30]. FMRP is located in the cytosol and the nucleus [31], as well as in dendrites, axons, and synapses [6,32,33]. FMRP expression changes over the course of brain development. It is expressed in embryonic brains, peaks in the first postnatal weeks (around day 7–10 in mice) and while the expression level reduces, it sustains a functionally significant level in adult neurons [34,35]. In addition to neuronal expression, FMRP is also identified in glial cells [36]. This broad expression profile likely reflects diverse functions of FMRP in cellular health. 

FMRP is best known as an RNA-binding protein that represses translation of a large subset of mRNAs, particularly mRNAs important in regulating synaptic development [33,37]. Besides mRNAs encoding synaptic and cytoskeletal proteins, FMRP regulates a number of voltage-gated ion channels either by regulating mRNA translation (canonical mechanism) or directly interacting with the ion channel protein (non-canonical mechanism), typically by modulating gating mechanics [4]. The loss of FMRP usually leads to reduced or enhanced function of these ion channels, which has repeatedly been linked to hyperexcitability particularly in cortical brain regions [4]. Interestingly, the FXS brain is characterized by dysregulation of ion channels that are not all FMRP targets and these are likely to also contribute to hyperexcitability [38,39]. These results suggest that the loss of FMRP directly and indirectly leads to hyperexcitability. In the following, we outline the different ways in which FMRP regulates voltage-gated ion channels. We refer the reader to previous reviews for a greater insight into the basic functions of ion channels [40]. We end this section with a discussion on the cell-type and region-specific effects of FMRP loss on ion channel functions and neuronal activity. The purpose of this chapter is to give the reader an overview of the mechanisms through which FMRP can regulate neuronal excitability. This information will be important for understanding how loss of FMRP leads to hyperexcitability. 

### 2.1. FMRP Interacts with mRNAs Encoding Voltage-Gated Ion Channels

In this section, we discuss how FMRP regulates the translation of a subset of voltage-gated ion channels, which can directly modify neuronal excitability and brain activity. The major take away is that FMRP regulates the translation of multiple ion channels, several of which have been found to contribute to neuronal hyperexcitability in FXS.

A comprehensive study identified multiple, direct interactions between FMRP and target mRNAs encoding for voltage-gated ion channels using high-throughput sequencing of RNA isolated by crosslinking immunoprecipitation (HITS-CLIP) technology [37]. In short, HITS-CLIP enables a large-scale identification of unknown protein-RNA interactions, in this case interactions between unknown mRNAs and FMRP. Among the identified interactions were the hyperpolarization activated cyclic nucleotide-gated cation channel 2 (gene: *HCN2*), potassium voltage-gated channel Kv4.2 (gene: *KCND2*), potassium voltage-gated channel Kv1.2 (gene: *KCNA2*), potassium large conductance calcium activated channel subfamily M, alpha member 1, a BK_alpha_ channel (gene: *KCNMA1*), potassium voltage-gated channel KQT like subfamily 2, Kv7.2 (gene: *KCNQ2*), calcium R-type (*CACNA1E*/Cav2.3), P/Q type (*CACNA1A*/Cav2.1) and five other *CACNx* channels, sodium channel type2, and alpha subunit (*SCN2A*/Nav1.2) ([37] see Table S3C–F). The Gene Ontology analysis identified the ‘synapse’, ‘dendrite’, and ‘neuron projection’ as cellular components enriched for mRNA targets of FMRP ([37] see Figure S3B). Although this study demonstrated direct interactions of FMRP with mRNAs encoding voltage-gated ion channels, very few studies have further investigated whether these RNA interactions led to altered protein levels and/or regulate ion channel surface expression and function. Instead, a multitude of studies have reported a set of distinct ion channels that are regulated by FMRP at the level of their mRNA. The incongruence between studies may be due to differences in sample types (e.g., species, in vivo or in vitro) or methodology. 

Prior to the HITS-CLIP analysis [37], the first evidence for a functional interaction of FMRP with an mRNA encoding an ion channel was for the potassium channel Kv3.1 [41]. Using RNA immunoprecipitation (RIP) in brain, a previous study identified FMRP association with *Kv3.1* mRNA [42], which was further shown to occur in brain stem enriched fractions. The K^+^ channel Kv3.1 is highly expressed in the lateral brain stem areas called medial nucleus of the trapezoid body (MNTB) and the anterior ventral cochlear nucleus (AVCN) that are both important for processing auditory input. In the MNTB, Kv3.1 protein is expressed in a tonotopic gradient, and Kv3.1 levels increase following auditory stimulation. This increase is necessary for proper auditory processing [41]. However, when Strumbos et al. assessed Kv3.1 expression in the *Fmr1* KO mouse brain stem areas, this tonotopicity was flattened, and Kv3.1 was not increased following acoustic stimulation [41]. The authors suggest that FMRP is important for establishing the Kv3.1 tonotopic gradient, and that this gradient is necessary for accurate auditory processing. This hypothesis supports the numerous reports finding impaired auditory processing in *Fmr1* KO mice and FXS patients [43,44]. Importantly, the Kv3.1 protein is expressed in a several cell types and brain areas and is associated with various (dys)functions including circadian rhythms and auditory seizures. Indeed, *Fmr1* KO and FXS patients are highly susceptible to auditory seizures and have disrupted circadian patterns [12,43,44,45,46,47]. FMRP could hold a particularly important role in regulating Kv3.1 tonotopicity as well as overall expression levels, potentially by both regulating mRNA translation and trafficking/stability [41]. 

HCN1 channel protein expression was significantly increased in hippocampal CA1 *Fmr1* KO dendrites, but not HCN2 channels [38]. This protein increase correlated with increased I_h_ current (referred to as the hyperpolarization-activated current) as well as an impairment in the activity-dependent regulation of the I_h_ current following either long-term potentiation or long-term depression paradigms. This study did not confirm that FMRP directly interacts with *Hcn1* mRNA, although it is likely that elevated protein expression resulted from loss of FMRP-mediated translation repression.

Two additional studies also identified an interaction between FMRP and the mRNA-encoding K^+^ channel mRNA Kv4.2 [48,49]. Gross et al. found that dendritic protein surface expression of Kv4.2 was reduced in CA1 *Fmr1* KO neurons, and that an mGluR5 antagonist could partially normalize this phenotype [48]. Thus, this study suggested a non-conventional role of FMRP in activating translation of Kv4.2 in hippocampal dendrites through untranslated sequences in the 3′ and 5′UTR, although the underlying mechanism is unclear. Other studies have also shown that while FMRP typically is a repressor, some FMRP targets are regulated in an opposite manner [50,51]. Later work has supported the idea that Kv4.2 ion channel expression is reduced by recording reduced K^+^-mediated A-currents in dendrites of CA1 hippocampal neurons in *Fmr1* KO mice [39,52], which are generally associated with Kv4.2 expression. In contrast, Lee et al. reported both increased levels of Kv4.2 in *Fmr1* KO hippocampal neurons, and that NMDA receptor signaling leads to an FMRP-dependent translation of Kv4.2, which is necessary for NMDAR-induced synaptic plasticity [49]. These findings are consistent with the canonical role of FMRP as a translational repressor, which is released during plasticity allowing for a short period of translation. FMRP did not appear to play a role in targeting and/or stabilizing Kv4.2 mRNA to the dendrites [49]. Thus, FMRP appears to regulate dendritic translation of Kv4.2 mRNA. Further research is needed to understand how FMRP might regulate translation of Kv4.2 bidirectionally or under different circumstances to regulate dendritic excitability and/or long-term plasticity [48,49]. 

As another example of FMRP mediated translational control impacting ion channel function, the regulation of Cav2.3 translation was shown to be dysregulated in *Fmr1* KO mice [53]. Using RIP, FMRP was shown to be associated with the mRNA encoding R-type voltage-gated calcium channel Cav2.3 in mouse brain. Hippocampal neurons from *Fmr1* KO mice have elevated Cav2.3 protein and larger R-currents under baseline conditions. Consistent with past work showing that FMRP represses translation downstream of gp1 metabotropic glutamate receptors [54], Cav2.3 protein levels as well as associated R-currents were both increased in response to gp1 mGluR stimulation in wild type neurons, but these responses were lost in *Fmr1* KO. These findings suggest that loss of canonical translational control by FMRP may contribute to aberrant calcium spiking, dendritic hyper-excitability, and seizures in FXS.

### 2.2. FMRP Directly Interacts with Ion Channel Proteins—Non-Canonical Mechanism

In this section, we discuss how FMRP can regulate ion channel proteins directly, a mechanism referred to as ‘non-canonical’. A direct protein-protein interaction endows FMRP with faster effects on neuronal excitability, enabling quick modifications of brain activity. The major take away is that FMRP can regulate neuronal excitability within seconds and minutes after a stimulus, which studies have linked to neuronal hyperexcitability.

Interestingly, studies reporting the non-canonical direct interaction between FMRP and ion channel proteins have been more abundant than those reporting FMRP’s canonical role in regulating ion channel mRNA translation (Figure 1). 

One of the first studies describing this phenomenon was conducted by Brown et al. They found that sodium-activated potassium channel, also called Slack, interacts with FMRP through its C-terminus [55]. In addition to the molecular interaction, they reported reduced Slack current in the lateral medial nucleus of the trapezoid body (MNTB), an area known for high Slack expression, in *Fmr1* KO mice. This study led the way to the hypothesis that FMRP may regulate gating of voltage-gated ion channel during sensory input, thereby regulating neuronal excitability and processing. 

Subsequently, several studies have shed light on the direct consequences of the absence of FMRP in regulating voltage-gated ion channel functions [56,57,58]. Deng et al. found that FMRP interacts with the auxiliary Beta4 subunit of the calcium-activated K^+^ channel BK and actually acts as a part of the calcium-sensing domain of the presynaptic BK channel [57]. During neuronal activity and neurotransmitter (NT) release, calcium levels increase in the presynaptic compartment. To ensure restricted NT release patterns/periods, the BK channel opens when presynaptic Ca^++^ levels reach a certain threshold, leading to an efflux of cations and thus hyperpolarization and cessation of NT release. Deng et al. found that the Beta4 subunit is necessary for opening the BK channel, and this opening depends on FMRP [57]. Indeed, BK currents are reduced in *Fmr1* KO CA3 neurons, leading to broader action potentials and prolonged NT released during a short-term plasticity paradigm of the CA3-CA1 circuit. These deficits were both translation- and mGluR5-independent. This study is particularly important because it provides a functional mechanism for the inability of FXS patients to filter out irrelevant sensory information. The majority of FXS patients suffer from sensory overload and major reductions in IQ. Lack of appropriate restriction of NT release during sensory input and plasticity will likely lead to unfiltered processing of every surrounding stimulus. Deng et al. used computational strategies to support this hypothesis, and they later found that a BK channel activator can rescue several phenotypes in the *Fmr1* KO mouse [59]. Excitingly, BK channel activators are already in clinical trials, but challenges regarding drug delivery in a cell-type- and brain area-specific manner have yet to be solved. Recent work from the same lab reported a direct interaction between FMRP and the gamma-aminobutyric acid-A (GABA_A_) receptor in hippocampal granule neurons [60]. FMRP regulates the open probability and conductance of GABA_A_ receptors with consequences for neuronal excitability and neuronal signaling processes important for learning and memory. These novel results may in part explain why GABA agonists improve behavior in the FXS mouse [61].

### 2.3. FMRP Regulates Ion Channels in a Cell Type and Region-Specific Manner

Our advancing understanding of FMRP’s role in regulating ion channels is exciting and has enormous therapeutic potential. However, we have realized that FMRP’s regulation of ion channels is highly dependent on the cell type and brain region [4,39]. Kalmbach et al. performed one of the most comprehensive studies of ion channel function in *Fmr1* KO mice across the layers and cell types of the cortex [39]. They found that some channels were dysregulated in opposite directions depending on the cell type and brain region. For example, the I_h_ current was downregulated in prefrontal L5 pyramidal neurons, while A-type K^+^ channels were increased in prefrontal pyramidal tract neurons [39]. These results are opposite to what was previously reported in the *Fmr1* KO hippocampus [38,48]. However, these ion channel dysregulations were not found in another subset of excitatory cortical neurons. These data provide an important warning of over-simplifying and generalizing phenotypes and suggest that an important focus of future research should be to understand possible convergent molecular mechanisms leading to diverse cell-type- and brain region-specific phenotypes.

## 3. Abnormal Excitability as a Neuronal Signature for FXS

FXS patients suffer from sensory hypersensitivity, sleep disturbances, and seizures [4]. Interestingly, these symptoms are associated with dysregulated neuronal excitability of cortical brain regions in FXS brains [12,45]. Likewise, children and adults with autism spectrum disorder (ASD) display dysregulated cortical excitability. The dysregulation is often towards hyperexcitability, i.e., increased activity of cortical brain regions [43,44,45,62,63]. Moreover, this cortical hyperexcitability has been linked to abnormal sensory processing and sensory hypersensitivity. In studies using electroencephalogram (EEG), FXS patients display enhanced N1 auditory responses, increased gamma power, and reduced ability to synchronize brain activity to sensory stimuli [43,44,45,62,63]. Thus, one of the main hypotheses in the field of FXS (and ASD) is the idea of a dysregulated excitatory/inhibitory balance causing abnormal sensory processing in cortical brain regions [1,2,4,64]. Restoring this balance continues to be a major focus for therapeutic development. 

The *Fmr1* KO mouse also expresses hyperexcitability in multiple cortical brain regions, including the auditory cortex, visual cortex, and somatosensory cortex both in vitro and in vivo. Some studies find unique features of hyperexcitability during certain developmental phases [65,66], and a large amount of studies support the idea of hyperexcitability in the mature *Fmr1* KO cortex [36,39,47,67]. Differences in the pattern or quality of the hyperexcitability across studies can potentially be attributed to mouse strain (i.e., genetic background of the *Fmr1* KO mouse) and technical approach (e.g., in vivo vs. in vitro, EEG vs. whole-cell patch clamp, and electrical recordings vs. imaging). Despite the incongruences across laboratories, the overall idea of cortical hyperexcitability in FXS remains strongly supported [4].

It is generally speculated that cortical hyperexcitability gives rise to dysregulated sensory responses, which can perturb sensory stimuli processing, sleep patterns and even become a susceptibility for seizure induction [2,4,64]. Seminal studies have reported a direct link between cortical hyperexcitability and impaired sensory processing. Especially interesting are the enhanced auditory responses which are found in people with FXS [44,45] as well as in the *Fmr1* KO mouse [68,69], which can be captured by EEG recordings in both humans and mice. In the KO, auditory stimuli lead to an abnormally long period of increased excitability in auditory cortex neurons, a defect which may lead to reduced habituation to auditory stimuli [68]. Indeed, sensory integration is proposed to be impaired in several sensory domains of FXS [14]. In the WT, selective deletion of *Fmr1* in excitatory auditory cortex neurons led to molecular, physiological, and behavioral phenotypes similar to the KO mouse in response to an auditory stimulus [70], suggesting an FMRP-specific role in regulating auditory processing and sensory integration. Human stem cell-derived cortical neurons from FXS patients display hyperexcitability, which is rescued by partial FMRP re-expression [71,72]. Thus, developing a drug to ameliorate cortical hyperexcitability could be a significant therapeutic advance for people with FXS and other people that suffer from sensory processing syndromes [14,73]. A current challenge is to consider combinatorial treatment approaches [74]. Other features of FXS, such as attention-deficit disorder and anxiety, may not be improved or even worsen by drugs that aim to reduce cortical hyperexcitability. This conundrum emphasizes the need for identifying specific molecular candidates for pharmacotherapeutic targets and for testing the effects of combinatorial drug treatments. 

It is important to mention that neuronal hyperexcitability also exists in other brain regions of the *Fmr1* KO mouse. Most notably, using in vitro whole-cell patch clamp recordings, hyperexcitability is reported in the basolateral amygdala (BLA) [75] and hippocampus (specifically the CA1 region) [57]. The BLA is important for fear and anxiety-like responses, which are elevated in the *Fmr1* KO mouse as well as FXS patients [76,77], and may be important for driving their augmented social withdrawal responses [76,78]. The hippocampus is important for learning and memory, and is thus of high interest in FXS due to its hallmark symptom of intellectual disability. Hippocampal-dependent learning/memory behaviors are impaired in *Fmr1* KO mice, and it has been suggested that hyperexcitability in the CA1 contributes to this deficit [57]. Clearly, dysregulated excitability is a phenotype that exists in several brain regions of FXS, but current research suggests that the mechanisms driving hyperexcitability and associated behavioral alterations are brain region specific. This notion is supported by several studies that can use pharmacological agents to rescue a set of phenotypes specific to one brain region, but not others [46,47,79]. Thus, investigating brain region-specific mechanisms underlying hyperexcitability in FXS is of utmost importance for advancing therapeutic strategies.

Recent studies have revealed novel mechanisms that may drive the onset of hyperexcitability in the cortex of FXS mice. Interestingly, not all of the identified mechanisms underlying the hyperexcitability are directly linked to the loss of FMRP. An outstanding question is therefore whether more general mechanisms exist to drive the concerted dysregulations of both FMRP-targeted and untargeted channels and receptors to drive hyperexcitability. The purpose of this chapter is to provide the reader with an overview of the published literature addressing the mechanisms driving cortical hyperexcitability in the FXS mouse model and FXS patients due to impairments in the function of metabotropic glutamate receptors, GABA receptors, or voltage-gated ion channels. Previous reviews have discussed these topics in depth, and we refer the reader to these for more information [4,27]. 

### 3.1. The Possible Contribution of Abnormal mGluR Signaling and Long-Term Plasticity to Cortical Hyper-Excitability 

The FXS field has in large part been dominated by the mGluR hypothesis: exaggerated mGluR signaling leads to enhanced protein synthesis-dependent removal of AMPA receptors causing sustained and amplified long-term depression (LTD), resulting in weakened synaptic strength possibly causing impaired learning and memory functions [80,81]. This hypothesis has repeatedly been confirmed in the hippocampus of FXS mice [46,79,82], but not in cortical areas [83]. Interestingly, LTP is absent in the Fmr1 KO cortex [84] While this review is not focused on the ample literature involving mGluR signaling, we briefly discuss the role of mGluR signaling in relation to cortical hyperexcitability. We refer the reader to other excellent reviews for further discussions on mGluR signaling in FXS [54].

Ronesi et al. found an increased association between the metabotropic glutamate 5 (mGlu5) receptor (mGluR5) and the short activity inducible isoform of the scaffolding protein Homer1a in *Fmr1* KO mice [47]. Homer1a, in contrast to long Homer isoforms, serves as an activator of molecules downstream of mGluR5 in response to synaptic stimulation, such as the Elongation Factor 2 Kinase (EF2K), which ultimately elevates neuronal activity. Indeed, deleting Homer1a expression restored cortical hyperexcitability, as well as seizures and elevated protein synthesis, but not hippocampal LTD [47]. Studies from our lab complemented this result by identifying increased levels of the PI3K enhancer PIKE in cortical *Fmr1* KO synapses, which activates PI3K/mTOR signaling, a pathway that is known to regulate neuronal activity [46,79]. Reducing PIKE expression in the *Fmr1* KO mouse normalized cortical hyperexcitability in an mGlur5-dependent but protein synthesis-independent manner [46,79]. The combined contribution of increased association with the short Homer1a isoform as well as increased PIKE levels likely serve as complementary pathology-inducing molecules that drive protein synthesis-dependent and -independent forms of cortical hyperexcitability. Lastly, the mGluR-triggered signaling cascade via the phospholipase C (PLC)-protein kinase C (PKC) pathway was also enhanced in the Enthorhinal Cortex of the FXS brain, which was directly linked to enhanced persistent Na^+^ current [67]. Reducing mGluR-PLC-PLC activity reduced the persistent Na^+^ current and normalized excitability levels. Interestingly, this phenomenon has previously been reported in other studies, providing further support for the idea that mGluR signaling can regulate neuronal excitability independently of long-term potentiation (LTP) and LTD [85].

In sum, mGluR5 downstream signaling is associated with cortical hyperexcitability in the FXS brain through protein-synthesis-dependent and -independent manners that are distinct from other brain regions (e.g., the hippocampus). When and how mGluR5 signaling becomes dysregulated in the FXS brain is currently unknown. 

### 3.2. GABA Receptor Signaling and Function in Cortical Hyper-Excitability

Given the important role for FMRP in regulating synaptic protein synthesis [27], surprisingly little evidence ties altered function of ligand gated receptors to cortical hyperexcitability [4]. However, the majority of the data that provide evidence to a ligand-gated receptor origin of hyperexcitability point to the inhibitory GABA_A_ receptors [4]. In this section we briefly discuss the documented relationship between GABA receptors and cortical hyperexcitability, a topic that other reviews have discussed extensively [86]. 

GABA_A_ receptor subunits are reduced at the mRNA and protein levels in the amygdala, cortex, hippocampus, and cerebellum of the *Fmr1* KO mice [61,87,88], and several studies report reduced GABAergic miniature inhibitory postsynaptic current amplitude and frequency supporting the hypothesis that reduced inhibitory tone could be one of the mechanisms driving neuronal hyperexcitability in FXS [75,76,87,88,89]. Studies in human FMRP KO neurons derived from embryonic stem cells also report reduced activation of GABAergic neurons with increased desensitization [90,91]. 

Interestingly, FMRP associates directly with GABA_A_ receptor mRNA, suggesting that FMRP functions as a translational activator for GABA_A_ mRNA [89], or, alternatively, FMRP may transport GABA_A_ receptor subunits into the membrane, accounting for the stark reduction in surface expression rather than total protein levels [88]. A recent study demonstrated that FMRP interacts directly with GABA_A_ receptors in hippocampal granule neurons [60], though more studies are needed to evaluate this interaction in other cell types. FMRP may also be important for the maturation of GABAergic neurons, with implications for brain development [92]. 

Gibson et al. revealed a significant reduction in the excitatory drive onto fast-spiking inhibitory neurons in layer 4 of acute sensory neocortex slices from *Fmr1* KO mice, but no difference in monosynaptic GABAergic activity onto neighboring neurons, suggesting that the origin of reduced inhibitory drive is due to defects in excitatory neurons and not GABAergic neurons [36]. Curiously, the inhibitory drive specifically by Somatostatin+ inhibitory neurons was enhanced in L2/3 cortical neurons [36], corroborating the idea that loss of FMRP, and other ASD-associated genes, triggers a compensatory increase in excitatory/inhibitory balance to stabilize synaptic activity [93,94], although this may not hold true for all brain regions [95]. Despite the local compensatory increases in GABAergic activity, the GABAergic defects have implications for local feedback inhibition in response to sensory input, represented as prolonged cortical circuit activity in both acute slices [36] and in vivo, a physiological phenotype associated with seizure susceptibility and sensory hypersensitivity [65,96]. Treatment with drugs that increase GABA_A_ receptor activity reduces anxiety responses in FXS mice as well as hyper-locomotion and sensory hypersensitivity [61]. These broad therapeutic benefits could be due to a general “re-balancing” of the excitatory/inhibitory balance. If the excitatory tone is abnormally elevated in FXS, increasing the inhibitory tone could ameliorate any hyperexcitability-mediated phenotypes regardless of the precise molecular mechanism underlying hyperexcitability [64].

### 3.3. Ion Channel Mechanisms Underlying Cortical Hyperexcitability

As already mentioned, the expression of several ion channels is altered in the *Fmr1* KO cortex via both canonical translation regulation and non-canonical protein interactions, which lead to abnormal excitability (See sections titled “Fragile X Mental Retardation Protein—more than just a translational regulator” and “Abnormal excitability as a neural signature for FXS”). However, not all of these expression differences are correlated with altered function or specifically cortical hyperexcitability. Unfortunately, the larger proportion of studies investigating the role of ion channels in hyperexcitability have been conducted in non-cortical regions such as the CA1 area of the hippocampus [38,52]. In this section, we highlight what is known to date regarding ion channel changes that contribute to hyperexcitability in FXS, specifically in the cortex.

As mentioned previously, FMRP interacts directly with the presynaptic auxiliary subunit Beta4, leading to opening of the K^+^ channel in response to Ca^++^ influx during action potential firing. The loss of this interaction leads to prolonged Ca^++^ influx and NT release, resulting in prolonged action potential firing [57]. The same group also discovered that persistent Na^+^ currents are increased in cortical *Fmr1* KO neurons [67] (Figure 2B). An increased persistent Na^+^ current reduced the action potential threshold and increased the number spikes fired by a cell. The increase in persistent Na^+^ current was mediated through an exaggerated mGluR5 signaling via the phospholipase C (PLC)-protein kinase C (PKC) pathway [67]. Inhibiting the persistent Na^+^ current normalized the abnormal action potential threshold and firing [67]. 

Zhang et al. identified a set of dendritic ion channels as the cause of cortical hyperexcitability and sensory hypersensitivity [69] (Figure 2B). In somatosensory (S1) cortex, they found a reduction in the dendritic HCN-channel mediated current I_h_ and HCN1 protein levels, as well as a reduction in the somatic Ca^++^ activated K^+^ channel “BK” current. These impairments contributed to the increased dendritic membrane resistance, increased action potential width, and increased action potential firing at the cellular level. When the authors pharmacologically activated the BK channel, they rescued dendritic hyperexcitability and sensory hypersensitivity in *Fmr1* KO mice in vivo. Overall, this comprehensive study suggested that synaptic integration and excitability are impaired due to altered ion channel expression in excitatory cortical neurons of the *Fmr1* KO [69].

Kalmbach et al. assessed excitability of different types of excitatory cortical neurons and found that unique ion channel changes in pyramidal tract projecting neurons led to their hyperexcitability [39]. In partial consistence with Zhang et al. [69], Kalmbach et al. found reduced I_h_ currents, increased A-type K^+^ channel function, and downregulated Kv1 currents [39]. Together, these alterations led to increased neuronal excitability by increasing membrane resistance and reducing fast and prolonged periods of hyperpolarization during periods of spiking (Figure 2). 

It is clear that a variety of changes in voltage-gated ion channels are directly related to cortical hyperexcitability in the *Fmr1* KO. This may raise the question as to whether pharmacological agents to normalize function of a single ion channel would be therapeutic. Zhang et al. provides encouraging data to suggest that despite multiple ion channel impairments, activating and thus restoring solely BK channel function significantly improved excitability and sensory responses in vivo [69]. These data are supported by a later study showing that genetic BK upregulation normalized epileptiform activity in *Fmr1* KO cortices in vitro [59]. However, the ion channel changes identified in the cortex are sometimes opposite of the changes identified in the hippocampus [39], and sometimes unique to the cortex [69]. Thus, increasing or decreasing the function of specific ion channels systemically would likely not be a therapeutic avenue due to extensive side effects and activation of compensatory mechanisms (see below). Instead, more work is needed to understand the mechanisms giving rise to ion channel alterations. Although some studies have connected the ion channel defects directly to the absence of FMRP [57], others have not [38,39,69]. Pharmacological modulation of general pathways/mechanisms regulating ion channel expression and function may provide a unique opportunity to normalize cortical hyperexcitability. 

In sum, cortical hyperexcitability in FXS is linked to a multitude of ligand-gated and voltage-gated channels, some of which are directly modulated by FMRP and others that are not. Although much progress has been made, it continues to be unknown whether specific channels and/or pathways cause hyperexcitability or if it is the concerted effort of indirect and direct channel dysfunctions that give rise to it and to associated symptoms (sensory hypersensitivity, seizures, sleep disturbances). If it is the latter, how does this mixed FMRP-specific and non-specific constellation of channel dysfunctions arise?

## 4. Homeostatic Intrinsic Plasticity as a Driver of Cortical Hyperexcitability in Fragile X Syndrome

One of the ideas emerging in the field of neurodevelopmental research is the importance of intact homeostatic plasticity for proper brain development and function [7,8,9,93,94,95,97]. Homeostatic plasticity ensures that neuronal activity levels are kept within a physiologically appropriate range certifying that synaptic strength or intrinsic excitability does not saturate and impair Hebbian plasticity [98,99]. Homeostatic plasticity can regulate neuronal activity through various mechanisms, such as modulating synaptic strength (called synaptic scaling) or neuronal excitability (called homeostatic intrinsic plasticity). Homeostatic plasticity has classically been studied in primary neuronal cultures and cultured brain slices, but more recent studies have demonstrated homeostatic plasticity in acute slices and in the living brain as well [98,100,101]. Both synaptic scaling and homeostatic intrinsic plasticity (HIP) have traditionally been triggered by long-lived (24–48 h) chemical drug treatments that either reduce or increase neuronal activity in cultured neurons [101,102,103,104]. Fascinatingly, both synaptic scaling and HIP are also triggered by long-term changes in sensory input, for example, loss of vision or reduced or enhanced whisker stimulation [81,105,106,107], emphasizing the translational nature of conducting research in simplified environments such as primary neuronal cultures. An abundance of research has and continues to investigate the molecular and physiological mechanisms underlying homeostatic plasticity, and we refer interested readers to previous reviews that comprehensively cover this topic [98,99]. In the following, we focus on the role of homeostatic plasticity in neurodevelopmental disorders.

Synaptic scaling has received the most attention in the context of brain function and development, and several studies have identified defects in this process in animal models of neurodevelopmental disease, for example, Schizophrenia, Rett Syndrome, and Fragile X Syndrome [8,9,108,109]. Dr. Chen’s lab was the first to identify impaired synaptic scaling in hippocampal neurons of a mouse model of Fragile X Syndrome [8,9]. Interestingly, other studies have reported intact synaptic scaling in cortical *Fmr1* KO neurons [7], corroborating the notion that FMRP loss has cell-type-specific effects on homeostatic plasticity function as well. Synaptic scaling is important for modulating synaptic strength and is hypothesized to play an important role in learning and memory, but it is unclear if impaired synaptic scaling could lead to hyperexcitability in FXS. Another type of homeostatic plasticity mechanism is homeostatic intrinsic plasticity (HIP), which regulates neuronal excitability [98,99]. HIP regulates a variety of voltage-gated ion channels at the axon initial segment and dendrites to maintain healthy neuronal activity levels. Interestingly, several of the ion channels regulated by HIP are also FMRP targets (Figure 3). Thus, loss of FMRP could directly impact HIP’s ability to adjust neuronal excitability throughout brain development. 

There are several reasons to believe that loss of FMRP may impair homeostatic intrinsic plasticity (HIP) function. First, although more studies are needed, HIP appears to be protein-synthesis-dependent [110,111] possibly involving FMRP regulation. Secondly, the FMRP-targeted ion channels clearly overlap with the ion channels reported to be regulated during HIP. Thus, the absence of FMRP could lead to exaggerated protein synthesis of FMRP-targeted ion channels upregulated during HIP [4] (Figure 3). Lastly, HIP is known to be important for structuring and regulating the function of developing circuits [112]. Previous studies have reported a developmental profile of hyperexcitability in the *Fmr1* KO mouse model, which is consistent with the relatively later onset of symptoms thought to be associated with cortical hyperexcitability in FXS patients. For example, seizures and sensory hyperarousal do not emerge until the second year of life [3]. Likewise, as the *Fmr1* KO mouse grows older, it becomes more likely to express behavioral and physiological phenotypes representing cortical hyperexcitability and sensory hypersensitivities [113,114]. 

Recent work from our lab identified a previously unknown abnormality in HIP function of cortical *Fmr1* KO neurons [7] (Figure 4). We identified abnormal HIP function in two distinct excitatory neuronal cell types of *Fmr1* KO neurons. One of these, the multi-spiking neurons became hyperexcitable following activity deprivation, whereas the other, the single-spiking neurons, displayed a failure of HIP. It is interesting to note the consistent cell-type specificity in various forms of plasticity when FMRP is lost [8,9,39]. As already mentioned, FMRP regulates several ion channels, multiple of which are also regulated by HIP. Indeed, we observed significant changes in the action potential waveform which indicated that fast and persistent Na^+^ channels as well as possibly slow K^+^ channels were dysregulated in *Fmr1* KO neurons during HIP [7]. Interestingly, both fast and persistent Na^+^ channels have previously been reported to be modulated by FMRP loss [52,67]. Preliminary data from our lab support the idea that Na^+^ currents are uniquely increased in KO neurons by HIP, supporting the hypothesis that HIP can drive the onset of hyperexcitability in FXS neurons (data not shown). It will be interesting for future studies to utilize drugs such as Riluzole to reduce persistent Na^+^ current function to normalize HIP-induced hyperexcitability. In the ion channel field, it is currently unclear which specific Na channel subunit carries the persistent Na current. A general idea is that increased phosphorylation of Nav1.2 and/or Nav1.6 subunits underlie the persistent Na^+^ current [115,116]. It is also possible that Na^+^ channel beta subunits mediate the persistent Na^+^ current by activating or inhibiting closing kinetics of the current. The development of better antibodies against phosphorylated and unphosphorylated Nav subunits will help elucidate the answers to these questions. 

FXS is best known as a disorder rooted in altered local protein synthesis [27], and dysregulated translation and protein stability has been reported in the dendrites, spines, and axons [33,117]. Yet, 48 h chemical drug treatments that elicit HIP and synaptic scaling do not lead to exaggerated upregulation of plasticity-induced proteins such as ion channels or ligand-gated receptors in *Fmr1* KO neurons [118]. However, FMRP targets diverse sets of mRNAs, several of which also regulate translation, protein degradation, and post-translational mechanisms [37]. For example, we know that FMRP regulates ubiquitination pathways that are likely important for expression of long-term depression [119]. Interestingly, a proteomics analysis from our lab identified greater changes in Gene Ontology modules associated with regulating phosphorylation [118]. Thus, we consider the possibility that loss of FMRP may lead to increased translation of phosphorylation and degradation machinery that ultimately drive complex changes in ion channel constellation and excitability during HIP. Although the above paragraphs mainly concern how HIP is dysregulated in multi-spiking *Fmr1* KO cells, we predict that similar mechanistic alterations exist in single-spiking *Fmr1* KO neurons. 

Our observation of multiple changes in ion channel function complicates the possibility of utilizing a single ion channel-targeting drug to modulate cortical hyperexcitability. One previous study demonstrated the positive effects of modulating the Ca^++^-activated K^+^ channel, BK, to reduce cortical hyperexcitability in *Fmr1* KO mice [59]. It is possible that altering the function of a single channel can dampen the overall activity, although not all dysfunctions are rescued. This is consistent with the relatively successful use of GABA-agonists to enhance the inhibitory tone of *Fmr1* KO neurons [77,89,120,121]. However, the side effects of altering the function of specific ion channels and GABA tone are significant and have halted clinical progress. It is likely that one of the major complications of these single-drug approaches is the cell-type specificity of FMRP across brain regions. To overcome this challenge, drugs should be engineered to target ion channel and/or ligand-gated receptor function in a brain region- and cell-type-specific manner. 

Our results comprise one of the first studies supporting the idea that abnormal homeostatic plasticity function can drive neuropathology. A study published a few months prior to ours demonstrated that a brief increase in neural activity led to greater neuronal silencing in a Drosophila model of Alzheimer’s disease (AD) [122]. In fact, the homeostatic synaptic silencing drove the progression of Abeta-induced neuronal silencing. Their study was one of the first to address the puzzling phenotypic change from neuronal hyperactivity in early disease stages to hypoactivity in later stages of AD. The idea that impaired homeostatic plasticity drives AD disease progression has previously been discussed [123]. More recently, Susco et al. used human-induced pluripotent stem cells (hiPSCs) to generate *Fmr1* KO excitatory neurons and reported age-dependent changes in intrinsic, but not synaptic, features that correlated with changes in neuronal excitability [124]. Their study supports the idea that HIP is of particular interest in understanding hyperexcitability in FXS and further leads our attention to the progressive nature of altered excitability. Recently, other ASD-associated genes have also been implicated with homeostatic plasticity function: loss of Shank3 expression in vitro and in vivo abolished HIP and synaptic scaling in visual cortex [97], suggesting that certain genes may be essential for homeostatic functions (e.g., Shank3) while other genes modulate plasticity expression (e.g., *Fmr1*). 

Research from Marder’s lab and others has identified the important role of HIP during central nervous system development [125]. Using computational approaches, their group identified that while many ion channels have redundant functions, a subset may be of particular relevance, which cannot be directly replaced when dysregulated or mutated [126]. Could the FMRP-regulated ion channels represent ion channels that are of particular importance in regulating membrane excitability during HIP? It is interesting to hypothesize that organisms with mutations leading to dysregulated ion channel composition may be able to compensate for these differences in the early parts of development (embryonic and early postnatal), but over time their parameter space becomes increasingly more restricted and eventually crashes. This way, HIP could be one of the driving forces leading the brain into a certain developmental pathway, which over time defines not only physiological activity levels, but also Hebbian types of neuronal plasticity and behavior. This hypothesis is in line with previous studies and reviews that suggest that altered activity-dependent processes during neuronal development are one of the major drivers of brain function impairment in FXS [127]. 

## 5. Ongoing and Future Directions

Research on cortical hyperexcitability in FXS continues to evolve, expanding beyond the initial theories limited to synaptic dysfunction. Several laboratories have now demonstrated a link between homeostatic plasticity and dysregulated excitability in mouse models of ASD [7,94,97]. In the following, we focus on the ongoing and future directions to uncover how loss of FMRP leads to cortical hyperexcitability through abnormal HIP. We discuss how FMRP’s direct and indirect effects on ion channel constellation may impair HIP function and how molecular pathways underlying HIP may be dysregulated in *Fmr1* KO neurons. Lastly, we discuss the need to test the developmental expression of HIP in vivo and the potential for critical periods of HIP’s influence on neuronal excitability and activity set point. 

In addition to these topics, it will also be important for research to understand how the insights from HIP in FXS could extrapolate to other forms of ASD. We already know that one other genetic-ASD mouse model, Shank3 KO, displays loss of HIP [97]. Do other ASD-related genes also lead to dysregulated HIP? Can we identify common loci of dysregulation? These insights could lead to common therapeutic strategies for a broad array of people with FXS, ASD, and other neurodevelopmental disorders. The purpose of this last chapter is to give the reader greater insights into where the field of FXS and hyperexcitability is developing. 

### 5.1. The Ion Channels Driving Cortical Hyperexcitability 

Loss of FMRP directly affects the function of a subset of ion channels, such as Slack and BK. In parallel, the loss of FMRP indirectly affects translation, trafficking, and function of many ion channels, such as voltage-gated Na^+^ channels. A major question is the degree to which these two different groups of ion channels contribute to cortical hyperexcitability in FXS. Do the FMRP-targeted ion channels prime neuronal hyperexcitability upon FMRP loss in the FXS brain? Does the loss of FMRP and consequent dysregulation of protein synthesis lead to an even greater impairment in ion channels that are not directly regulated by FMRP? When does each of these groups of ion channels start contributing to cortical hyperexcitability? Addressing these questions will inform therapeutic strategies on pharmaceutical targets and timing of interventions. From the perspective of homeostatic plasticity, we speculate that loss of FMRP leads to an initial dysregulation of neuronal excitability (due to impaired expression of FMRP-targeted ion channels, such as BK, Slack, Kv4.2 and HCN2) [48,49,55,57], but that this can be homeostatically compensated for through other non-FMRP-targeted ion channels. This idea is supported by research demonstrating that neurons can achieve the same excitability phenotype through a variety of unique ion channel constellations [125,126] and that early in development *Fmr1* KO neurons often do not display hyperexcitable phenotypes [7,113,128]. However, as the brain continues to develop and receives increasingly more sensory input, a larger repertoire of ion channels is necessary to adjust excitability and maintain stable neuronal activity levels. In the FXS brain, initial reduction in available ion channels (due to impaired regulation of FMRP-targeted channels) may lead to an inability to properly compensate for changes in neuronal activity levels. This hypothesis may address why certain FXS symptoms, such as sensory hypersensitivity and seizures, are not observed until the second or third year or life [3]. 

Intriguing work has demonstrated how some ion channels are critical for HIP to regulate excitability and maintain stable activity levels [126,129], and other papers have pointed out how the expression (and thus functionality) of certain ion channels is directly regulated by the conductance of other ion channels [130,131]. This array of research brings forth the idea that cortical hyperexcitability may be a consequence of abnormal ion channel constellation and function in the FXS brain that restricts the capacity for homeostatic adjustments. Future work could test whether restoration of just the FMRP-targeted ion channels in early development would circumvent the trajectory of cortical hyperexcitability. 

### 5.2. The Molecular Pathways Driving Cortical Hyperexcitability 

Numerous studies have reported impaired cellular signaling due to FMRP loss causing impaired plasticity, such as mGluR-LTD and other forms of synaptic plasticity [79,132]. However, the molecular pathways underlying HIP are relatively unknown, with no specific role of FMRP yet revealed. A major question is whether HIP drives cortical hyperexcitability not (only) due to abnormal ion channel constellation but due to impairments in the molecular pathways that mediate HIP. 

Richard Baines’ lab published one of the first studies dissecting the molecular pathways underlying HIP [110,111]. Using drosophila motoneurons, his group identified that expression of the transcription factor Pumilio correlated with changes in synaptic activity: increased synaptic excitation reduced Pumilio whereas reduced synaptic excitation increased it. Changes in Pumilio expression triggered changes in voltage-gated Na^+^ channel expression with consequences for neuronal excitability (note that drosophila only have one Pumilio paralog, whereas mammals have two: PUM1 and PUM2). Subsequent studies have demonstrated a similar role of PUM2 in regulating the voltage-gated Na^+^ channel Nav1.6 in rat visual cortex [133]. Interestingly, crosslinking studies have found RNA-dependent interactions between both Pumilio paralogs (PUM1/2) and FMRP [134]. FMRP and PUM1/2 bind to each other’s mRNA targets suggesting that PUM1/2 and FMRP may co-regulate subsets of mRNAs. Loss of FMRP could thus impair regulation of PUM1/2′s mRNA targets [134], perhaps with consequences for voltage-gated Na^+^ channel expression and function. It is important to mention that FMRP is particularly important during phases of activity-dependent plasticity, whether that be due to intrinsic changes over development or in response to a learning paradigm. It is therefore to be expected that the consequences of FMRP loss are most noticeable in response to these events, which has been repeatedly demonstrated in studies finding no significant changes in neuronal excitability/activity at baseline yet pronounced differences in response to activity-dependent plasticity such as LTD or HIP [7,81]. When developing therapeutic strategies, it is imperative that we focus on targeting activity-dependent processes that are particularly sensitive to FMRP loss. 

A recent publication uncovered a necessary role of mitochondrial signaling in regulating neuronal activity setpoints [135]. The neuronal activity setpoint defines a target range that homeostatic plasticity ensures neuronal activity is kept within [98,99]. HIP and synaptic scaling are two of the homeostatic mechanisms that return neuronal activity to this setpoint upon long-term changes in activity levels, for example due to a drug treatment. If we understand what determines the neuronal setpoint, it may also give insights into how homeostatic mechanisms, such as HIP, are triggered. Recent work from our lab identified compartmentalized changes in mitochondrial morphology and function in response to 48 h treatment with the inhibitory cocktail of tetrodotoxin (TTX) and (2R)-amino-5-phosphonovaleric acid (APV) [136]. TTX blocks voltage-gate sodium channels while APV inhibits NMDA receptor function, resulting in a significant reduction of action potential firing and spontaneous NMDA receptor signaling. This treatment is a classic trigger of homeostatic plasticity [103,104], and mimics events during brain development characterized by significant drops in neuronal activity (for example when the NT GABA switches from excitatory to inhibitory) [92]. Following TTX/APV treatment in WT neurons, mitochondria residing in dendrites became larger, a phenotype that has been hypothesized to correlate with increased translation of mitochondrial proteins and increased ATP output [137,138]. In contrast, mitochondria residing in the axon initial segment maintained their original size but displayed a significant reduction in their membrane potential (25). Reduced mitochondrial membrane potentials typically signify increased workload and/or reduced mitochondrial health [139,140]. Current work in our lab is establishing how changes in mitochondrial membrane potential in the AIS may mediate HIP-induced translation and insertion of ion channels to regulate neuronal excitability [136]. Overall, these results support the idea that mitochondria may be important for triggering homeostatic plasticity mechanisms to ensure that a certain activity setpoint is maintained [135,136]. Given the exaggerated HIP phenotype in *Fmr1* KO neurons, we hypothesized that these activity-dependent changes in mitochondrial morphology and membrane potential would be similarly exaggerated. Surprisingly, *Fmr1* KO neurons displayed no changes in mitochondrial shape or function after TTX/APV treatment [136]. Rather, *Fmr1* KO neurons already displayed increased dendritic mitochondrial morphology and reduced mitochondrial membrane potential in the AIS at baseline, supporting previously stated ideas that *Fmr1* KO neurons are primed for plasticity responses at the molecular level, leading to exaggerated changes in neuronal physiology upon activity-dependent changes [132]. Why and how these molecular differences are not revealed at the physiological level at baseline remains puzzling but may be explained by partially intact homeostatic compensatory mechanisms when there is no additional duress on the system. Understanding the molecular pathways underlying mitochondrial plasticity during HIP may provide insights for future therapeutic strategies to correct cortical hyperexcitability in FXS. 

### 5.3. New Cellular Models

While the FMRP mouse model has been used extensively in the past two decades with interesting and potentially important insight into molecular mechanisms of FX syndrome, there is an urging need for expanding these observations to the human realm. Two reasons underlie this urge; first, the mouse model deals with a situation where the FMRP gene is knocked out right from the early stages of development, whereas in the human case, the mutation is developing gradually. The functional implication of this difference is not entirely clear. Second, in the human FXS is a major illness, and the patient is unable to function in most human attributes (e.g., communication), and the proposed treatments failed thus far to ameliorate the disease symptoms. In contrast, the FXS mouse is far less affected by the disease. One approach that employs human tissue was developed recently and involves human embryonic stem cells, either native or inducible pluripotent stem cells (IPS). In these cells, grown in dissociated cultures, there is easy access to the neurons across different developmental stages [90,91,141]. The monitoring of spontaneous network as well as synaptic activity in these neurons allow detection of several properties of the network, which could be employed for a correlation between molecular and functional properties of the neurons. These and similar in-vitro systems are expected to enhance our knowledge of FXS and propose novel therapeutic approaches.

### 5.4. Determine How Neuronal Excitability Is Expressed during Brain Development In Vivo and If HIP Function Is Altered at Some or All Developmental Time Points

Activity deprivation led to abnormal excitability via HIP in *Fmr1* KO neurons, and this supports the idea that HIP contributes to the expression of dysregulated excitability in *Fmr1* KO neurons. An important future study is to evaluate whether abnormal HIP is also displayed in vivo of *Fmr1* KO brains. Moreover, it will be important to test the developmental trajectory of excitability and HIP expression in *Fmr1* KO brains. 

A common approach to induce homeostatic plasticity in living mice is by trimming one side/region of their whiskers for days or even weeks [93,106,142]. This trimming leads to compensatory responses in the contralateral barrel cortex, demonstrated by a recovery of activity measured through multi-unit electrodes. The perturbation can also be conducted by removing visual input and evaluating visual cortex activity [100]. To specifically study in vivo cellular excitability, one must perform whole-cell patch clamping in vivo. Previous studies have used in vivo whole-cell recording to measure intrinsic excitability of neurons during homeostatic recovery from whisker trimming [106,142], and have thus demonstrated the feasibility of this approach. *Fmr1* KO mice and FXS patients present with tactile hypersensitivity, which is thought to be mediated by neuronal hyperexcitability [4]. The barrel cortex is therefore an optimal region for addressing how HIP is expressed and potentially abnormal in the living *Fmr1* KO mouse. It will be of interest to investigate if *Fmr1* KO neurons demonstrate cell-type-specific responses to whisker trimming, leading to hyperexcitability in one subset of excitatory neurons, and loss of HIP in another subset. The idea of cell-type-specific plasticity responses in subsets of excitatory neurons in vivo is supported by multiple studies showing that Regular Spiking (RS) and Intrinsic Bursting (IB) excitatory neurons, recorded in vivo, display significantly different temporal and functional responses during whisker trimming [142]. While IB neurons display a fast recovery of activity, the RS neurons are much slower. Moreover, responses to stimulation of spared whiskers changes during the homeostatic recovery in a cell-type-specific manner, such that RS cells associated with one of the spared whiskers show reduced evoked responses, while IB cells overcompensate and become even more active in response to whisker stimulation. It is likely that these cell type specific responses rely on different types of homeostatic mechanisms [106,142]. 

An intriguing and unanswered question is whether homeostatic plasticity mechanisms, including HIP, have critical periods. One could hypothesize that HIP/synaptic scaling are most important during early postnatal development where the brain is overwhelmed with an astounding number of intrinsic and extrinsic challenges. To test this hypothesis, one would need to perform multi-unit electrode recordings in vivo at varying developmental time points during activity perturbations. Although this will be challenging in young mice, recent studies have demonstrated that recording from even P0 mice is feasible [143]. Subsequent studies can continue with whole-cell recordings to assess whether HIP and scaling undergo critical periods at the same time point. 

The proposed studies outlined above will be critical for our understanding of normal brain development and the role of homeostatic plasticity in the process. If homeostatic mechanisms display critical periods, this could suggest that they have particularly significant effects on brain structure and function during this period, and we hypothesize that restoring HIP during this time period would have defining consequences for *Fmr1* KO brain development. Overall, these studies will clarify the role of homeostatic plasticity mechanisms in the living *Fmr1* KO brain and further qualify this type of plasticity as a therapeutic target. In light of the genetic differences between mice and humans, it will be important to also test homeostatic plasticity function in hiPSC derived neurons/organoids. Dr. Lu Chen’s group found that synaptic upscaling was impaired in hippocampal *Fmr1* KO neurons [8,9], and this phenotype was replicated in hiPSC-derived neurons [144], supporting the idea that homeostatic plasticity alterations translate from mice to humans. Restoring homeostatic plasticity, specifically HIP, during early brain development of FXS would likely have wide-spread therapeutic effects on brain circuitry and function. 

## Figures and Tables

**Figure 1 ijms-23-06315-f001:**
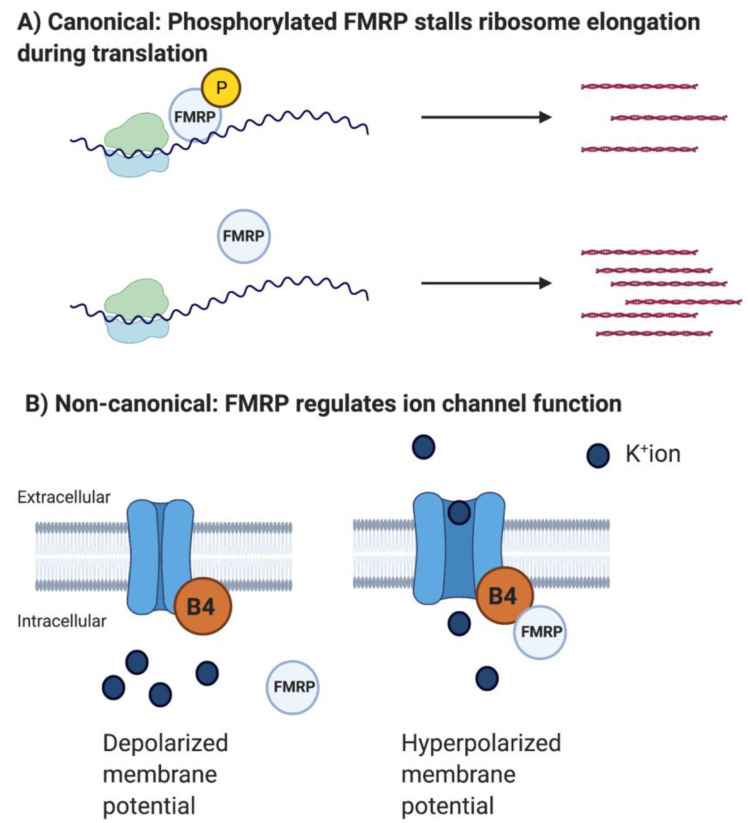
FMRP’s Canonical and Non-Canonical Mechanisms. (**A**) FMRP stalls the elongation of translation (Darnell et al., 2011). Dephosphorylation of FMRP leads to removal of translational repression, which may allow ribosome elongation and increases total protein synthesis, also shown in Figure 1. (**B**) FMRP regulates protein function directly. Example: FMRP associates with Beta4 (B4) subunits causing opening of the Ca^++^-activated K^+^ BK channel. See Richter et al., 2015 for an in-depth discussion of FMRP’s canonical and non-canonical mechanisms.

**Figure 2 ijms-23-06315-f002:**
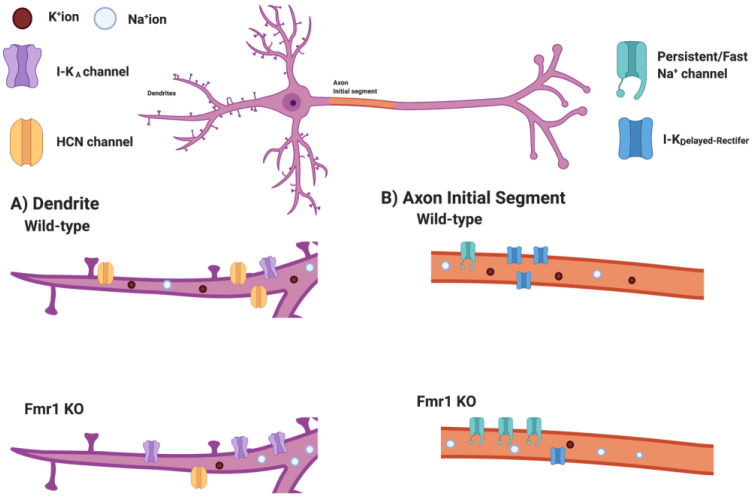
Ion channel dysregulations that contribute to cortical hyperexcitability in *Fmr1* KO neurons. (**A**) *Fmr1* KO dendrites present with altered composition of ion channels when measured with electrophysiological and immunofluorescence measures. Increased I-K_A_ currents (potentially mediated by Kv4.2 channels) can support faster firing of dendritic action potentials. Reduced HCN currents (likely through HCN1 channels) reduce dendritic hyperpolarization, thus disinhibiting dendritic action potential firing. (**B**) *Fmr1* KO axons display increased currents through persistent and fast Na^+^ channels, both of which are associated with neuronal hyperexcitability. The increase in Na^+^ currents is combined with reduced delayed rectifier K^+^ currents, further enhancing a hyperexcitable state of the *Fmr1* KO axon.

**Figure 3 ijms-23-06315-f003:**
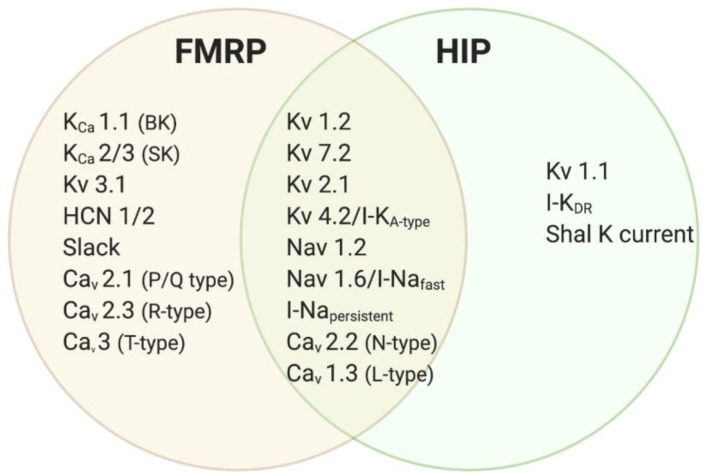
FMRP and Homeostatic Intrinsic Plasticity regulate distinct and overlapping groups of ion channels. Venn diagram of ion channels that are regulated by FMRP or HIP, and the subset that overlap (middle line). The ion channel regulation can be either at the level of mRNA or protein function. Note that the FMRP ion channel targets continue to be validated depending on cell type, age, and brain region. For other reviews on ion channels targeted by FMRP (see Darnell et al., 2011; Contractor et al., 2015; Brager and Johnston, 2014).

**Figure 4 ijms-23-06315-f004:**
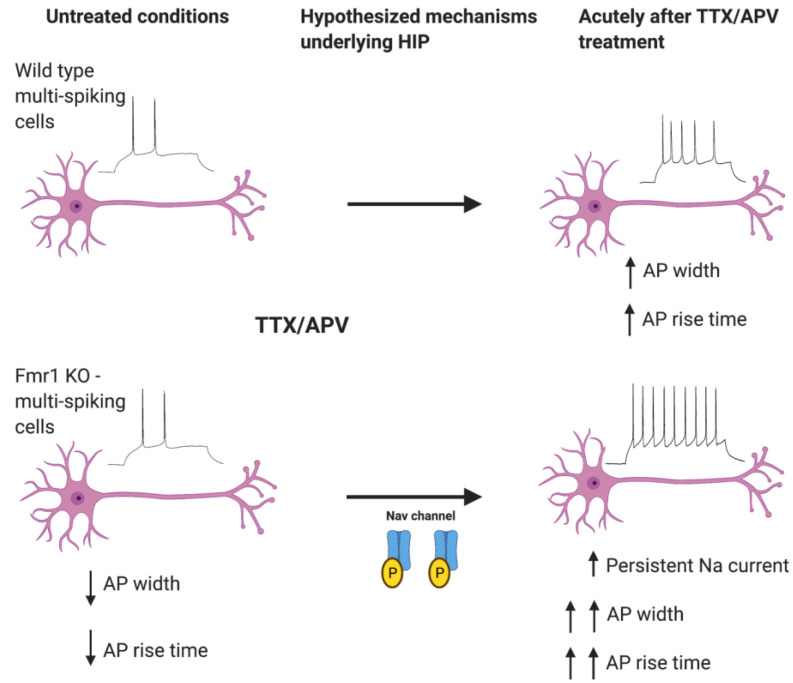
Activity deprivation triggered hyperexcitability in multi-spiking *Fmr1* KO neurons. Activity deprivation led to exaggerated HIP in multi-spiking *Fmr1* KO neurons, which resulted in a hyperexcitable phenotype compared with treated WT neurons. The excitability changes correlated with changes in the action potential waveform, which were changed to a greater extent in the *Fmr1* KO. We hypothesize that increased phosphorylation of Nav channels contributes to exaggerated expression of HIP in *Fmr1* KO neurons.

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
