# Peer review of "Mechanisms Driving the Emergence of Neuronal Hyperexcitability in Fragile X Syndrome"

_ijms, 2022, doi:10.3390/ijms23116315_

Round 1

Reviewer 1 Report

Review paper written by Bülow and colleagues summarizes current knowledge on mechanisms that are involved in the development of brain hyperexcitability in patients with FXS. It is a great peace of work that can be further improved. Main criticism - the review is too broad, authors do not pay enough attention to highlight importance of their own studies. This makes the review a bit boring, as readers do not understand what is new in the review, where the original data of the authors, how the individual parts of the review are interconnected.

Major

There is a need to give in the first chapter the definition of hyperexcitability, describe major ion channel regulators as well as channels involved in the regulation HIP.

The review would benefit if authors highlight importance of their own studies (most likely it is homeostatic regulation of synaptic plasticity) in the abstract and at first chapter of the review.

The readers would benefit if authors would navigate them through the text via explaining why they write each chapter and how information written in each chapter would help to understand hyperexcitability observed in the FXS neurons.

Minor

Lines 126, 249 – dot should be moved closer to corresponding number“.  FMRP”

Line 171 – please decipher the abbreviation “Ih” current

Line 269 – term ASD has to be deciphered

Line 350 – there is no reference in the reference list that is matched to Kazdoba et al, 2014

Author Response

Please note that all the changes within the document are highlighted in yellow for convenience.

Reviewer 1:

Major requests:

  • Request to provide definition of hyperexcitability and mention ion channel regulators and channels involved in HIP
    • We have reworded sentence 53-54 to clearly define neuronal hyperexcitability
  • Request to highlight the importance of the authors’ own studies in the first chapter and the abstract.
    • We kindly thank you for this request. We have added more direct references to the work from our lab in the abstract and the first chapter.
  • Request to better navigate reader through the document.
    • We have included meta-descriptions of the purpose and content of relevant sections to provide a clearer understanding for the reader why the section/chapter is relevant to understand hyperexcitability in FXS.

Minor requests: We have implemented all of the suggested edits. We thank you for your stark attention to detail!

Reviewer 2 Report

This is an excellent review that offers an authoritative update on the pathophysiology underlying neuronal hyperexcitability in FXS. Definitely, the most interesting part is that covered by section 4 and part of section 5 (that related to ongoing directions). To emphasize the contents of these sections I suggest that the authors shorten other sections. I think that a reduction in size would render the review more readable.. Some specific points that can be improved:

  1. Line 57: “severe seizures (including epilepsy)”. I think it is the other way around: It is epilepsy that includes seizures. Moreover, on line 70 it is stated that “people with FXS also suffer from epilepsy (typically representing benign Focal Epilepsy of Childhood (BFEC)”. Benign is opposite to severe. In my own experience, epilepsy is not a major issue in FXS. Therefore I would tune down the concept that neuronal hyperexcitability leads to severe seizures in FXS.
  2. Lines 88-89: there is a redundancy to be corrected.
  3. Line 127: HITS-CLIP. Please explain.
  4. Lines 200-207. Why in bold?
  5. Line 221. I suggest that the authors provide at least one specific reference for the statement “FMRP stalls the elongation of ”
  6. “developing a drug to ameliorate cortical hyperexcitability could be a significant therapeutic advance for people with FXS” I would be cautious on this point, given that the psycho-stimulating drug methylphenidate is largely used in FXS. The authors may wish to add a brief comment on this point.
  7. Line 563: Figure 3 is somehow incongruent in placing vis-a-vis a protein (FMRP) and a function (HIP). I wonder whether the authors could resolve this incongruence.
  8. Line 771: TTX/APV. Please explain.
  9. Line 778: “Future work in our lab is establishing” should be changed to “Current work in our lab is establishing”
  10. One last point: The review deals extensively with synaptic function but never mentions the altered morphology and number of dendritic spines in FXS. A brief mention of this phenomenon would be appropriate.

Author Response

Please note that all the changes within the document are highlighted in yellow for convenience.

Reviewer 2:

  1. Request to reduce focus on seizures in FXS
    1. Our statements are informed from research with FXS mice and patients, for example from Dr. Berry-Kravis’ lab. According to recent meta-reviews, around 8-16% of males with FXS suffer from seizures, some of which are diagnosed as epilepsy. On your request, we have removed “severe” as an adjective associated with seizures for with people with FXS in our review.
    2. Hagerman et al., 2017, Nature Reviews 2017, https://doi.org/10.1038/nrdp.2017.65. Citation from review: “Seizures occur in ∼8–16% of males and 3–7% of females with FXS, typically present in the first 5 years of life, and are the most substantial medical problem for children with FXS”
  2. Request to solve redundancy
    1. This is now fixed
  3. Request to explain HITS-CLIP
    1. We have included a definition of the HITS-CLIP technique
  4. Request to remove bold letters
    1. We do not see the letters written in bold.
  5. Request to include reference
    1. We have included a reference (Darnell et al., 2011) in the figure legend that specifically states “FMRP stalls the elongation of”
  6. Request to include comment on the effects of other drugs
    1. This is an excellent point, and we have included a brief discussion on the importance of testing the effects of combinatorial treatment approaches
  7. Request to address an incongruence between FMRP and HIP in figure 3
    1. Figure 3 represents the distinct and overlapping ion channels regulated by HIP and FMRP. While HIP is a form of plasticity and FMRP a protein, FMRP has direct implications for ion channels functioning (just like HIP). It is still unknown whether FMRP is a part of the HIP process, but literature suggests that it might. We hope this explanation will resolve the perceived incongruence.
  8. Request to explain TTX/APV
    1. We have now included a definition of each of the drugs
  9. Request to change “Future” to “Current”
    1. We have implemented the change
  10. Request to mention spine morphology and number abnormalities in FXS
    1. We have included a new section (section 1.3) that briefly comments on the findings on spine defects in FXS patients and mice.